# Monitoring the Efficacy of High-Flow Nasal Cannula Oxygen Therapy in Patients with Acute Hypoxemic Respiratory Failure in the General Respiratory Ward: A Prospective Observational Study

**DOI:** 10.3390/biomedicines11113067

**Published:** 2023-11-16

**Authors:** Zhanqi Zhao, Mei-Yun Chang, Tingting Zhang, Chien-Hung Gow

**Affiliations:** 1School of Biomedical Engineering, Guangzhou Medical University, Guangzhou 511436, China; 2Department of Critical Care Medicine, Peking Union Medical College Hospital, Chinese Academy of Medical Sciences, Beijing 100730, China; 3Institute of Technical Medicine, Furtwangen University, 78054 Villingen-Schwenningen, Germany; 4Department of Internal Medicine, Far Eastern Memorial Hospital, New Taipei City 22060, Taiwan; changmy30@yahoo.com.tw; 5Department of Biomedical Engineering, College of Medicine, Kyung Hee University, Seoul 02447, Republic of Korea; zttysu@khu.ac.kr; 6Department of Internal Medicine, Changhua Hospital, Ministry of Health and Welfare, Changhua 513007, Taiwan; 7Department of Healthcare Information and Management, Ming-Chuan University, Taoyuan 33348, Taiwan

**Keywords:** acute hypoxemic respiratory failure, electrical impedance tomography (EIT), high-flow nasal cannula (HFNC), neutrophil-to-lymphocyte ratio (NRL), regional ventilation distribution

## Abstract

High-flow nasal cannula (HFNC) is widely used to treat hypoxemic respiratory failure. The effectiveness of HFNC treatment and the methods for monitoring its efficacy in the general ward remain unclear. This prospective observational study enrolled 42 patients who had acute hypoxemic respiratory failure requiring HFNC oxygen therapy in the general adult respiratory ward. The primary outcome was the all-cause in-hospital mortality. Secondary outcomes included the association between initial blood test results and HFNC outcomes. Regional ventilation distributions were monitored in 24 patients using electrical impedance tomography (EIT) after HFNC initiation. Patients with successful HFNC treatment had better in-hospital survival (94%) compared to those with failed HFNC treatment (0%, *p* < 0.001). Neutrophil-to-lymphocyte ratios of ≥9 were more common in patients with failed HFNC (70%) compared to those with successful HFNC (52%, *p* = 0.070), and these patients had shorter hospital survival rates after HFNC treatment (*p* = 0.046, Tarone-Ware test). Patients with successful HFNC treatment had a more central ventilation distribution compared to those with failed HFNC treatment (*p* < 0.05). Similarly, patients who survived HFNC treatment had a more central distribution compared to those who did not survive (*p* < 0.001). We concluded that HFNC in the general respiratory ward may be a potential rescue therapy for patients with respiratory failure. EIT can potentially monitor patients receiving HFNC therapy.

## 1. Introduction

Patients with lung diseases often experience hypoxemia and respiratory distress in general adult respiratory wards. Oxygen therapy is the first-line treatment for these patients. To mitigate hypoxemia, oxygen can be provided through conventional oxygen therapy, including general nasal cannulas, simple face masks, adjustable aerosol masks, and non-rebreathing masks. Non-invasive or invasive ventilators are required for patients with impending respiratory failure. Clinically, appropriate selection of the oxygen delivery device, fraction, and flow is required according to the condition of individuals and the severity of hypoxemia over time. However, the support provided by conventional oxygen therapy may be insufficient for several patients in respiratory wards. The intolerance and discomfort associated with ventilation may also cause treatment failure. The use of a high-flow nasal cannula (HFNC) can reduce the respiratory rate and work of breathing in post-operative or intensive care unit (ICU) patients [1,2].

Over the last 10 years, interest in using HFNC as a first-line treatment has increased in the adult population [3]. HFNC is widely used to treat hypoxemic respiratory failure. The arterial partial pressure of O_2_ and fraction of inspired oxygen (PaO_2_/FiO_2_) ratio are significantly better with HFNC compared with general oxygen therapy [4,5]. Most studies on the effects of HFNC on respiratory failure have focused on ICU or post-operative patients. The effectiveness of HFNC treatment and the methods for monitoring its efficacy in patients with acute hypoxemic respiratory failure (AHRF) in the general ward remain unclear.

This study aimed to assess the efficacy of HFNC therapy in patients with AHRF. We initially investigated the association between initial blood test results and HFNC outcomes. Subsequently, we monitored regional ventilation distributions using electrical impedance tomography (EIT) after initiating HFNC. EIT is a monitoring instrument that analyzes boundary voltage–current data on the chest wall surface during breathing, enabling clinicians to measure and observe the dynamic ventilation distribution and regional lung perfusion [6]. EIT has previously been used to evaluate the effects of HFNC on infants with bronchiolitis [7]. However, the correlation of EIT patterns in patients receiving HFNC therapy remains to be seen, with limited data available. The information gathered in this study may aid in determining whether clinical factors and EIT data predict HFNC treatment outcomes in patients with AHRF in the general respiratory ward.

## 2. Materials and Methods

### 2.1. Ethics Statement and Patients

This prospective observational study was approved by Far Eastern Memorial Hospital (FEMH) in Taiwan (FEMH-107139-F). The patients provided written informed consent to participate in this study. Patients with AHRF admitted in the general respiratory ward from April 2019 to December 2021 were screened. The inclusion criteria were as follows: (1) reported dyspnea without oxygen support, (2) age between 20 and 90 years, (3) meeting the AHRF criteria (SpO_2_ < 91% or PaO_2_ < 60 mmHg without oxygen support, PaO_2_/FiO_2_ ratio < 300 with oxygen support), and (4) suitable for HFNC therapy as per the attending physician’s discretion. The exclusion criteria included pregnancy, brain injury, epilepsy, myocardial infarction, and missing informed consent (patients who did not agree to participate).

### 2.2. High-Flow Nasal Cannula Therapy and Measurements

The patients included in this study received HFNC therapy using Precision Flow (Vapotherm, Exeter, UK), with initial flow settings of 40 L/min, temperature of 37 °C, and FiO_2_ of 0.5, or AIRVO2 (Fisher & Paykel Healthcare, Auckland, New Zealand), with initial settings of flow of 60 L/min, temperature of 37 °C, and FiO_2_ of 0.5. The FiO_2_ and flow rate were adjusted to maintain the SpO_2_ between 92% and 98%. Patient demographics were also recorded at baseline. The comfort level was evaluated using a 7-item questionnaire with a scale of 0 to 10 (from 0, no discomfort, to 10, totally intolerable). Failure of HFNC treatment was defined as death during HFNC treatment, the necessity of masked bilevel positive airway pressure or endotracheal tube intubation, and mechanical ventilator support to maintain ventilation.

Electrical impedance tomography (EIT) was routinely performed on patients with HFNC unless contraindications for EIT were present (e.g., pacemaker, automatic implantable cardioverter defibrillator, implantable pumps, large wound on the chest) or patients refused to use EIT. A 16-electrode EIT electrode belt was placed on the chest at the fifth intercostal space, with a reference electrode positioned on the abdomen (PulmoVista 500; Dräger Medical, Lübeck, Germany). An alternating current was applied during the sequential rotation. The frequency and amplitude of the current were automatically determined based on the background noise in the measurement environment. The surface potential difference between the adjacent electrode pairs was measured and recorded at a frequency of 20 Hz. EIT was performed with the patients spontaneously breathing in the supine position at three time points: T0, 30 min before the start of HFNC; T1, 2 h after HFNC initiation; and T2, 24 h after HFNC initiation. A 15-min measurement was recorded at each time point. Suction and positional changes were avoided during the EIT measurements. The images were reconstructed using the manufacturer’s software (EIT Data Review Tool version 6.3; Dräger Medical, Lübeck, Germany). Custom software programmed in MATLAB R2015 (MathWorks Inc., Natick, MA, USA) was used for the offline analysis of the EIT data.

### 2.3. Electrical Impedance Tomography Data Analysis

Functional EIT (fEIT) tidal variation (TV) was derived by computing the difference between the end-expiration and end-inspiration images, capturing the variation during tidal breathing. Tidal images of 1 min were averaged to increase the signal-to-noise ratio:(1)$${TV}_{i}=\frac{1}{N}\sum_{n=1}^{N}\left({∆Z}_{i,Ins,n}-{∆Z}_{i,Exp,n}\right)$$
where *TV_i_* is the pixel *i* in the fEIT image, *N* is the number of breaths within the analyzed period, and Δ*Z_i,Ins_* and ΔZ*_i,Exp_* are the pixel values in the raw EIT image at the end inspiration and end expiration, respectively. When *TV_i_* was <0, a value of 0 was assigned to *TV_i_*.

Three EIT-derived indices were investigated and assessed to quantify the spatial and temporal distributions of ventilation. The global inhomogeneity (GI) index was computed from the tidal EIT images to characterize the variability in ventilation [8]:(2)GI=∑l∈lungTVl−Median(TVlung)∑l∈lungTVl

*TV* represents the differential impedance value in the tidal images, *TV_l_* signifies the pixel within the identified lung area, and pixel *l* is classified as part of the lung region if *TV_l_* is >10% × max (TV). *TV_lung_* encompasses all the pixels depicting the lung region. A heightened GI indicates substantial variation among the impedance values of the tidal pixels. The center of ventilation (CoV) illustrates the distribution of ventilation affected by factors such as gravity or different lung conditions (weighted relative impedance values based on anteroposterior coordinates) [9]:(3)CoV=∑(yi×TVi)∑TVi×100%
where *TV_i_* is the impedance change in the fEIT image for pixel *i*, y*_i_* is the height of pixel *i*, and the value is scaled such that the bottom of the image (dorsal) is 100% and the top (ventral) is 0%.

The tidal image was divided into four horizontal and anterior-to-posterior segments of equal height (regions of interest (ROIs)). The ventilation distributions in these regions were calculated and are denoted as ROIs 1–4.

The regional ventilation delay (RVD) index describes the regional delay in ventilation by comparing the rising time of the pixel impedance to the global impedance curve [10], which can be used to assess the tidal recruitment and derecruitment:(4)RVDl=tl,40%Tinspiration,global×100%
where t_*l*,40%_ is the time required for pixel *l* to reach 40% of its maximum inspiratory impedance change and *T_inspiration,global_* denotes the inspiration time calculated from the global impedance curve.

### 2.4. Statistical Analyses

Data analysis was performed using MATLAB R2015 (MathWorks Inc., Natick, MA, USA). Clinical data were analyzed based on categorical variables, which were compared using the chi-squared test. Fisher’s exact test was applied when the expected value was <5. In the hospital ward, survival was calculated using the Kaplan–Meier method. Differences in the survival curves were measured using the log-rank test or Tarone-Ware test. Whether HFNC success or failure was associated with demographics, FiO_2_, flow rate, or blood cell count was also explored. The Lilliefors test was used to test for normality. For normally distributed data, results were expressed as mean ± standard deviation. For non-normally distributed data, the results were presented as medians (minimum–maximum). A two-way analysis of variance was used to evaluate the differences between the three time points and HFNC outcomes. *p* < 0.05 was considered statistically significant.

## 3. Results

Forty-two patients were enrolled in this study, and EIT measurements were available for twenty-four patients. The clinical characteristics of all patients are listed in Table 1.

The primary cause of acute hypoxemic respiratory disease was pneumonia, followed by obstructive lung disease and advanced lung cancer. No patient with interstitial lung disease was included in the study subjects. Ten patients failed HFNC treatment (10 of 42, 24%). Among them, an advanced lung cancer patient who suffered from pulmonary lymphangitic carcinomatosis with a do-not-intubate order was treated with HFNC oxygen therapy for four days but failed. Subsequently, she received non-invasive mechanical ventilator support and passed away after one day. Another nine patients who failed HFNC treatment had pneumonia as the primary cause, including one patient who progressed to acute respiratory distress syndrome and received endotracheal intubation, mechanical ventilator support, and ICU admission; five patients received non-invasive ventilation (masked bilevel positive airway pressure) support; two patients had do-not-intubate orders and passed away; and one patient had a hospital-acquired coronavirus disease 2019 (COVID-19) and was under non-rebreathing mask support and eventually died.

The remaining 32 (76%) patients were successfully weaned off HFNC, and their oxygen demand was reduced to a simple mask, nasal cannula, or room air oxygenation after their disease became stable. Twenty-four patients with pneumonia, five patients with obstructive lung diseases, and three patients with advanced lung cancer discontinued HFNC and converted to conventional oxygen therapy after proper treatments. Among the lung cancer cases, two patients with malignant pleural effusion successfully removed HFNC after pigtail catheter drainages and treatment with albumin plus diuretics; one patient who experienced hemoptysis and aspiration was stabilized after medical treatment and weaned off HFNC. The overall in-hospital mortality rate was 29% (12 of 42). Two patients successfully received HFNC treatment during the AHRF episode but died later due to other causes.

The potential prognostic factors associated with HFNC success or failure were analyzed (Table 2). Patients in the HFNC success group had better in-hospital survival (survival rate, 94%, 30 of 32) compared with those in the HFNC failure group (survival rate, 0%, 0 of 10; *p* < 0.001). Other factors were similar between the two groups. Oxygen delivery was also analyzed in the HFNC success and failure groups. The delivered FiO_2_ levels were divided into low (<60%) and high (≥60%). Oxygen flow was divided into low (<40 L/min) and high (≥40 L/min). The selected cut-off FiO_2_ and flow rates were based on an increased risk of intubation among critical patients treated with HFNC [11]. Patients were divided into low- and high-FiO_2_ groups, and flow rates were delivered on Day 1 of HFNC. No significant differences were observed between the success and failure groups. Other cut-off points for FiO_2_ (40 and 50%) and flow rates (20 and 30 L/min) were also explored. Similar results were obtained.

Complete blood cell count, differential count, venous blood gas (VBG), and biochemical examination (blood urea nitrogen (BUN), creatinine, and albumin) results were analyzed. We observed that patients in the HFNC treatment failure group had a higher frequency of anemia (Day 1 hemoglobin (Hgb) < 9 mg/dL), with 50% (5 of 10), compared with those in the success group (16%, 5 of 17; *p* = 0.040). Leukocytosis and platelet count were not prognostic factors. The neutrophil-to-lymphocyte ratio (NLR), which assesses inflammatory or infectious conditions and represents physiological stress, was also evaluated. We categorized NLR levels into two groups, ≥9 and <9, as NLR ≥ 9 has been associated with predicting mortality in critically ill patients with pneumonia [12]. Although a trend was observed, no significant difference was found between the two groups. Other factors, such as pH value (acidosis vs. normal vs. alkalosis), CO_2_ level in the VBG, BUN level, creatinine value, and albumin level, were not different between the two groups.

Most patients tolerated HFNC with limited discomfort during body turnover and movement (Table 3). Univariate analyses of prognostic factors for in-hospital survival were performed using the Kaplan–Meier method and the log-rank test or the Tarone-Ware test (Table 4). HFNC treatment failure was associated with poorer survival outcomes for both ward survival (*p* < 0.001, Figure 1a) and survival after HFNC treatment (*p* < 0.001, Figure 1b). Similar to the prognostic factors associated with HFNC success or failure, age, sex, smoking status, and major diseases were not predicting factors for in-hospital survival either for the ward or after HFNC treatment. The delivered FiO_2_, flow rate, Hgb level, WBC count, or platelet count could not separate the survival outcomes. Patients with NLRs ≥9 were associated with shorter in-hospital survival (39 days) compared with those with NLRs <9 (110 days), but the difference was not significant (*p* = 0.062 using the log-rank test, Figure 1c). Similar results were observed for hospital survival after HFNC treatment (31 vs. 41 days, NLR ≥ 9 vs. <9, *p* = 0.055 using the log-rank test and *p* = 0.046 using the Tarone-Ware test, Figure 1d).

The EIT data were available for 24 patients, including 18 patients with pneumonia, 4 patients with obstructive lung disease, and 2 patients with advanced lung cancer. Among them, 20 patients were successfully treated with HFNC, and 4 failed. Regarding in-hospital survival, 18 patients remained alive, while 6 patients died. All EIT data retrieved from patients who failed HFNC treatment or died were from those who had pneumonia as the primary cause of AHFC. No significant differences were found in the EIT-based parameters of GI or RVD among the three time points or between the HFNC success and failure groups. At T1 and T2, the ventilation distribution of patients with successful HFNC treatment was more significant towards the center (CoV closer to 50%). The data were significantly different between HFNC success and failure at T0, T1, and T2 (*p* < 0.05, Figure 2a). Similarly, at T1 and T2, patients who survived HFNC treatment had a more central distribution compared with those who did not survive (*p* < 0.001, Figure 2b).

## 4. Discussion

In this prospective observational study, we demonstrated that HFNC therapy was clinically applicable for managing patients with acute hypoxemic respiratory failure in a general respiratory ward. The success rate of weaning off HFNC was 76%. The in-hospital mortality rate was 29%. EIT can potentially help monitor patients receiving HFNC therapy.

HFNC can be beneficial, feasible, and safe for patients with AHRF in the general ward, including those with lung cancer, chronic obstructive pulmonary disease, and mild-to-moderate adult respiratory distress syndrome [13]. Recently, a randomized multicenter clinical trial study in the ICU demonstrated that HFNC treatment could improve AHRF compared with conventional oxygen therapy or non-invasive ventilators and that the 90-day survival rate was better among patients with HFNC compared with those receiving conventional oxygen therapy and non-invasive ventilation [14]. Several studies have described the early predictors of HFNC outcomes in AHRF. Clinical factors, such as baseline heart rate, alveolar–arterial PO_2_ difference, Sequential Organ Failure Assessment score, and vasopressor use, are significantly higher in HFNC failure groups than in HFNC success groups among patients with specific diseases admitted to the ICU [15,16,17]. These results suggest that the initial disease severity and organ dysfunction may be good predictors of HFNC failure in patients with acute respiratory failure. In the present study, we found that NLRs > 9 could possibly predict HFNC treatment failure in the general respiratory ward. The NLR is considered a potential marker for predicting the requirement of a high-flow oxygen nasal cannula and invasive mechanical ventilation in acute hypoxemic respiratory cases [18,19,20]. A high NLR in early AHRF indicates the severity of the disease and organ dysfunction and has been observed in several conditions associated with tissue damage-induced systemic inflammatory response syndrome [21]. Although several studies and our data have demonstrated that HFNC treatment failure certainly has a higher NLR [18,20,22], the cut-off NLR value to determine HFNC treatment outcomes requires further examination. Other factors, such as pH, CO_2_ levels in VBG, BUN, creatinine, and albumin levels, were not different between the two groups.

Patients in the failure group developed anemia more frequently than those in the successful group (*p* = 0.040). Research reporting the relationship between anemia and outcomes of HFNC oxygen therapy in AHRF patients is rare. However, previous studies have shown that anemia patients with hypoxemic respiratory failure have poor treatment outcomes. Anemia patients in the ICU had adverse outcomes and a higher extubation failure rate [23,24]. Patients who survived acute respiratory distress syndrome with anemia at ICU discharge were associated with worsened exercise capacity and more dependency for activities of daily living [25]. Keng et al. reported an increased risk of mechanical ventilator weaning failure among patients with anemia and poor oxygenation at respiratory care center admission [26]. Reade et al. demonstrated that anemia patients (Hgb < 10) are associated with 90-day mortality in cases of hospitalized community-acquired pneumonia [27]. Recently, a large multicenter cohort study reported that COVID-19 patients with anemia are associated with disease severity and mortality [28]. A possible mechanism for the more frequent occurrence of anemia is that inhaled oxygen from the environment crosses the alveolar–capillary membrane into the bloodstream. Most oxygen is bound to Hgb in the red blood cells, although a small amount dissolves in the plasma. Oxygen is then transported from the lungs to the peripheral tissues, where it is removed from the blood and used to promote aerobic cellular metabolism. As the percentage of subjects with Hgb levels of <9 was higher in the HFNC failure group compared with the HFNC success group, the concentration of Hgb in these subjects was likely low, and their ability to carry oxygen became inadequate. Consequently, the hypoxemia of the corresponding subjects did not improve during HFNC therapy, and higher-level support was required.

EIT enables clinicians to measure and observe dynamic regional ventilation distribution [6]. Basile et al. reported reduced GI values in patients receiving HFNC, indicating an improvement in uneven ventilation distribution after HFNC therapy [29]. Pérez-Terán et al. demonstrated that HFNC significantly decreased the respiratory rate and increased end-expiratory lung impedance in patients with respiratory diseases [30]. Recently, Li et al. reported that ventilation distributions among patients with acute respiratory failure during their first hour in the ICU were slightly different but were insignificant in predicting HFNC failure [31]. An effort has been made to use machine learning methods to predict HFNC outcomes using EIT [32]. However, the number of participants was limited to those with effective machine learning. The clinical application of EIT in general respiratory wards has rarely been reported. We found that EIT could be used to monitor patients receiving HFNC therapy in the general ward. Patients with successful HFNC treatment had a more central distribution 2 h after HFNC treatment, lasting for at least 24 h (Figure 2a). Several possible mechanisms may explain this finding. First, HFNC has a low positive end-expiratory pressure effect (e.g., 5 cm H_2_O). This may promote ventilation redistribution within a short period in cases of AHRF. Second, HFNC improves patient comfort and provides a stable oxygen concentration under a continuous, high oxygen flow with adequate humidity [4,33]. In the present study, we observed that ventilation in the failure group was distributed slightly towards the ventral region (Figure 2a). Previous studies have suggested that ventilation distribution during spontaneous breathing may be related to diaphragmatic activity [34,35]. The differences found in the CoV might indicate decreased respiratory effort in the failure group, implying possible fatigue of the respiratory muscles. Other EIT-based parameters, such as RVD, did not differ between the subgroups. We suspect that during spontaneous breathing, the inspiratory time is significantly short to obtain stable RVD values. A recent study showed a high coefficient of variation for RVD among healthy subjects [36], which may explain why no significant differences were observed in the present study.

HFNC treatment is generally comfortable for patients in AHRF. In a previous study, clinical staff reported easy use of HFNC devices, whereas patients reported relatively high comfort levels while breathing humidified and preheated air [37]. The benefits of patient tolerance and more reliable FiO_2_ delivery due to dead space flushing make HFNC an excellent method for oxygen delivery. Early initiation of HFNC reduces inspiratory effort, thereby reducing pulmonary transvascular pressure and protecting the lungs from patient-inflicted lung damage [38]. A previous study applied the therapeutic benefits of HFNC, namely, the tolerance of long ventilation times, reduced nursing workload, and significant reduction in 90-day mortality previously described in the literature in favor of HFNC, for acute hypoxemic respiration and compared the results with other forms of non-invasive ventilation (e.g., continuous positive airway pressure or bilevel positive airway pressure) [14]. In addition, HFNC therapy improves the respiratory rates, tolerance, and comfort of interstitial lung disease patients with acute hypoxemic respiratory failure who receive supportive care [39].

Our study has some limitations. First, the clinical data were collected from a single medical center. Second, although this was a prospective study, the sample size was relatively small. Third, the causes of the diseases were heterogeneous, which may have caused statistical insignificance in some of the investigated parameters. Further large-scale studies focusing on a single disease for these prognostic factors are necessary to validate these clinical findings.

## 5. Conclusions

This study suggests that HFNC therapy in general respiratory wards may be a potential rescue therapy for patients with respiratory failure. EIT potentially monitors patients receiving HFNC therapy.

## Figures and Tables

**Figure 1 biomedicines-11-03067-f001:**
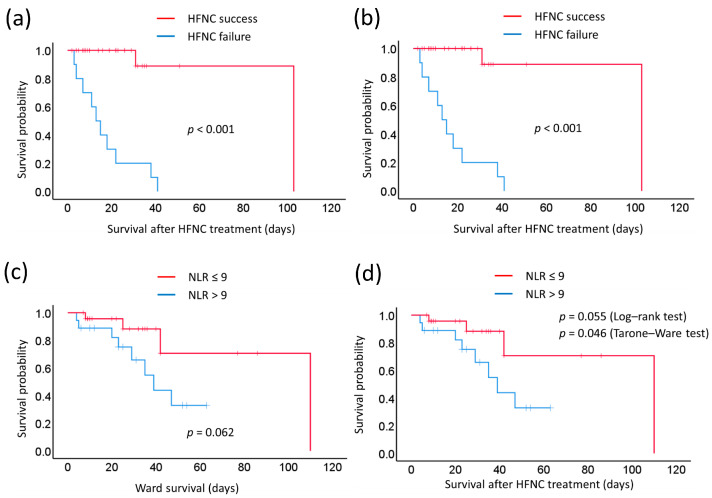
Kaplan–Meier curves of in-hospital survival for acute hypoxemic respiratory failure under HFNC oxygen therapy: (**a**) The ‘ward survival’ of patients with HFNC success or HFNC failure. (**b**) The ‘survival after HFNC treatment’ of patients with HFNC success or failure. (**c**) The ‘ward survival’ of patients with high neutrophil-to-lymphocyte ratios (NLRs) of ≥9 or low NLRs (NLR < 9). (**d**) The ‘survival after HFNC treatment’ of patients with high NLRs of ≥9 or low NLRs of <9. (*n* = 42). HFNC, high-flow nasal cannula.

**Figure 2 biomedicines-11-03067-f002:**
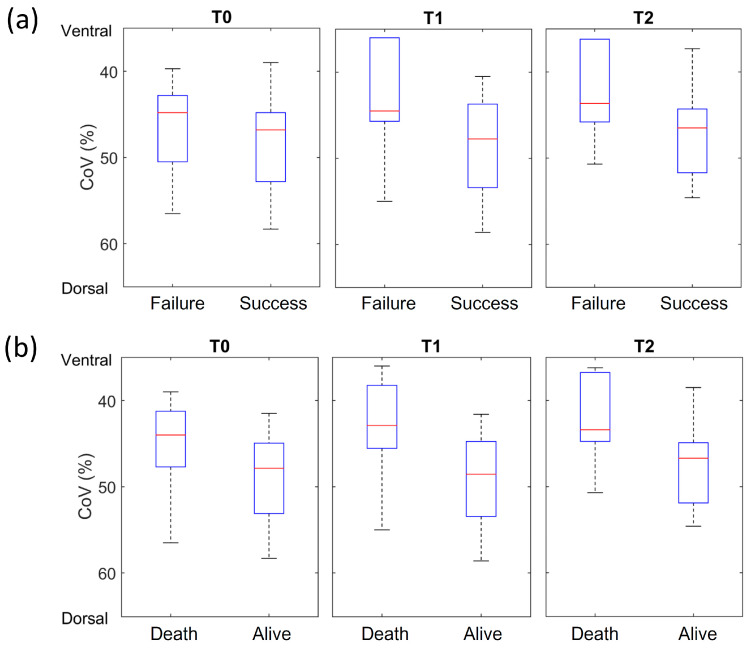
EIT-derived CoV at different times: (**a**) between the HFNC success and failure groups and (**b**) between the survival and death groups. T0, 30 min before HFNC; T1, 2 h after HFNC initiation; and T2, 24 h after HFNC initiation. (*n* = 24). EIT, electrical impedance tomography; HNFC, high-flow nasal cannula.

**Table 1 biomedicines-11-03067-t001:** Clinical characteristics of patients receiving HFNC therapy for acute hypoxemic respiratory failure.

Clinical Characteristic	Values
Patients, *n*	42
Age, years	
Median (minimum–maximum)	74.5 (52–88)
Sex, *n* (%)	
M	24 (57)
F	18 (43)
Smoking status, *n* (%)	
Current/Ever	23 (55)
Never	19 (45)
Primary cause of respiratory failure, *n* (%)	
Pneumonia	33 (78)
Obstructive lung diseases	5 (12)
Lung cancer	4 (10)
Heart failure, *n* (%)	
Yes	3 (7)
No	39 (93)
HFNC efficacy, *n* (%)	
Success	32 (76)
Failure	10 (24)
Event (days ± SD)	
Hospital admission	29.4 ± 23.1
HFNC to discharge	20.8 ± 17.8
HFNC use	11.3 ± 8.9
Survival/Death, *n* (%)	
Survival	30 (71)
Death	12 (29)

Abbreviations: F, female; HFNC, high-flow nasal cannula; M, male; *n*, number; SD, standard deviation.

**Table 2 biomedicines-11-03067-t002:** Factors associated with HFNC therapy treatment outcomes in patients with acute hypoxemic respiratory failure.

Factor	HFNC Success (*n* = 32)	HFNC Failure (*n* = 10)	*p*-Value
Age, *n*			1.000
<75	16	5	
≥75	16	5	
Sex, *n*			1.000
M	18	6	
F	14	4	
Smoking status, *n*			0.729
Current/Ever	18	5	
Never	14	5	
Survival/Death, *n*			<0.001 *
Survival	30	0	
Death	2	10	
Primary cause of respiratory failure, *n*			0.564
Pneumonia	24	9	
Obstructive lung disease	5	0	
Lung cancer	3	1	
Heart failure, *n*			1.000
Yes	3	0	
No	29	10	
HFNC, FiO_2_ on Day 1 (%), *n*			0.268
<60	21	4	
≥60	11	6	
HFNC, flow on Day 1 (L/min), *n*			0.466
<40	21	8	
≥40	11	2	
Hemoglobin (mg/dL), *n*			0.040 *
≥9	27	5	
<9	5	5	
White cell count (per dL), *n*			1.000
≥12,000	12	4	
<12,000	20	6	
Platelets (per dL), *n*			1.000
≥80,000	29	9	
<80,000	3	1	
Neutrophil-to-lymphocyte ratio, *n*			0.070
≥9	11	7	
<9	21	3	
pH level, *n*			1.000
≤7.34	4	1	
7.35–7.45	17	6	
≥7.46	11	3	
CO_2_ level (mmHg), *n*			0.625
<40	13	4	
=40–55	13	7	
≥55	6	3	
BUN (mg/dL) on Day 1, *n*			0.719
≤25	15	6	
>25	17	4	
Creatinine (mg/dL) on Day 1, *n*			0.660
≤1.3	26	7	
>1.3	6	3	
Albumin (mg/dL) on Day 1, *n*			0.128
<3.0	9	6	
≥3.0	23	4	

Abbreviations: F, female; HFNC, high-flow nasal cannula; M, male; *n*, number. *p*-values were calculated using a two-sided chi-squared test. * Statistically significant values (*p* < 0.05).

**Table 3 biomedicines-11-03067-t003:** Seven-item questionnaires to evaluate the comfort level of high-flow nasal cannula oxygen therapy for patients with acute hypoxemic respiratory failure.

Item	Questionnaire	Score ± SD
1	How comfortable is your nose or face when using oxygen equipment?	2.0 ± 1.5
2	How comfortable is your mouth/nose/throat (whether it is dry) when using oxygen equipment?	2.4 ± 1.8
3	How comfortable do you feel to swallow when using oxygen equipment?	2.2 ± 1.6
4	How comfortable do you feel during eating when using oxygen equipment?	2.4 ± 2.0
5	What extent do you feel that the use of oxygen equipment affects coughing?	2.4 ± 1.8
6	How much do you feel about body turnover when using oxygen equipment?	3.8 ± 1.9
7	How much do you feel about movement and activity using oxygen equipment?	3.9 ± 2.0

We recorded scores from Day 1 to Day 3 and expressed them as an average ± standard deviation (SD); the scores were defined as follows: 0 (no discomfort), 3 (slightly uncomfortable but acceptable), 5 (not very comfortable), 7 (uncomfortable and unbearable), and 10 (totally intolerable).

**Table 4 biomedicines-11-03067-t004:** Prognostic factors for in-hospital survival of patients with acute hypoxemic respiratory failure under HFNC therapy.

		Ward Survival		HFNC to Discharge	
Factor	Patients *n*	Median (Days)	*p*-Value	Median (Days)	*p*-Value
Age, years			0.705		0.473
<75	21	47		41	
≥75	21	110		103	
Sex			0.346		0.574
M	24	42		41	
F	18	110		38	
Smoking status			0.599		0.850
Current/Ever	23	42		41	
Never	19	110		38	
HFNC efficacy			<0.001		<0.001
Success	32	23		103	
Failure	10	110		13	
Primary cause of respiratory failure			0.890		0.716
Pneumonia	33	47		38	
Non-pneumonia	9	47		41	
HFNC, FiO_2_ on Day 1 (%)			0.136		0.282
<60	25	110		103	
≥60	17	42		38	
HFNC, flow on Day 1 (L/min)			0.948		0.760
<40	28	47		38	
≥40	14	110		103	
Hemoglobin (mg/dL)			0.356		0.380
≥9	32	NR		38	
<9	10	47		41	
White cell count (per dL)			0.973		0.358
≥12,000	16	NR		NR	
<12,000	26	47		41	
Platelets (per dL)			0.894		0.577
≥80,000	38	110		103	
<80,000	4	47		38	
Neutrophil-to-lymphocyte ratio			0.062		
≥9	18	39		31	0.055
<9	24	110		41	0.046 * (T–W)
pH level			0.928		0.776
7.35–7.45	23	47		38	
Abnormal	19	NR		NR	
CO_2_ (mmHg) level			0.733		0.290
40–55	19	47		38	
Abnormal	23	110		41	
BUN (mg/dL) on Day 1			0.522		0.526
≤25	21	42		38	
>25	21	110		103	
Creatinine (mg/dL) on Day 1			0.357		0.540
≤1.3	33	47		41	
>1.3	9	29		31	
Albumin (mg/dL) on Day 1			0.493		0.387
<3.0	15	47		38	
≥3.0	27	110		41	

Abbreviations: F, female; HFNC, high-flow nasal cannula; M, male; *n*, number. *p*-values were calculated using the Kaplan–Meier method and using the log-rank or Tarone-Ware test (T–W test) to measure all differences in survival. * Statistically significant values (*p* < 0.05).

## Data Availability

All data generated or analyzed in this study are included in the published article. Data are available upon reasonable request to the corresponding author.

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
