# Peer review of "Monitoring the Efficacy of High-Flow Nasal Cannula Oxygen Therapy in Patients with Acute Hypoxemic Respiratory Failure in the General Respiratory Ward: A Prospective Observational Study"

_biomedicines, 2023, doi:10.3390/biomedicines11113067_

Round 1

Reviewer 1 Report

Comments and Suggestions for Authors

This study eventually suggests that HFNC therapy in general respiratory wards may be a potential salvage therapy for patients with respiratory failure. 

A few aspects need to be better clarified

1 Which patients were the ones included in the failure group? What type of disease?

2 Based on failure due to anemia, it would be better to show other studies or data that may better explain this phenomenon that could be of interest in daily practice 

3 Why there are no inclusive and exclusive criteria?

4 Finally, the authors asserted that the causes of the diseases were heterogeneous, which may have caused false statistical significance in some of the parameters investigated. This is why I find that at this point we still need to carry on a discussion that allows us to bypass this deficit...

5 I would highlight more the failure rather than the success considering the the small number of patients included in this study

Comments on the Quality of English Language

minor editing needed

Author Response

Reviewers’ comments

Review Report, Reviewer 1

This study eventually suggests that HFNC therapy in general respiratory wards may be a potential salvage therapy for patients with respiratory failure.

A few aspects need to be better clarified

  1. Which patients were the ones included in the failure group? What type of disease?

Response: Thanks for the reviewer’s important question. We explained the details regarding the HFNC failure group on page 5 of the revised manuscript: “Among them, an advanced lung cancer patient suffered from pulmonary lymphangitic carcinomatosis with a do-not-intubate order treated with HFNC oxygen therapy for four days but failed. Subsequently, she received non-invasive mechanical ventilator support and passed away after one day. Another 9 patients who failed HFNC had pneumonia as the primary cause, including 1 patient who progressed to acute respiratory distress syndrome and received endotracheal intubation, mechanical ventilator support, and ICU admission; 5 patients received non-invasive ventilation (mask bilevel positive airway pressure) support; 2 patients with a do-not-intubate order and passed away; and 1 patient had a hospital-acquired coronavirus disease 2019 (COVID-19) and was under non-rebreathing mask support and eventually died.”

       On page 9 of the revised manuscript, we explained the type of diseases in patients with EIT available data: The EIT data were available for 24 patients, including 18 patients with pneumonia, 4 patients with obstructive lung disease, and 2 patients with advanced lung cancer. Among them, 20 patients were successfully treated with HFNC, and four failed. Regarding in-hospital survival, 18 patients were alive, while six patients died. All EIT data retrieved from patients who failed HFNC treatment or died were those who had pneumonia as the primary cause of AHFC.”

2 Based on failure due to anemia, it would be better to show other studies or data that may better explain this phenomenon that could be of interest in daily practice.

Response: Thanks for the reviewer’s essential suggestion. We added the following paragraph to discuss the association of anemia and acute hypoxemic respiratory failure on page 11 of the revised manuscript: “Research reporting the relationship between anemia and outcomes of HFNC oxygen therapy in AHRF patients is rare. However, previous studies showed that anemia patients with hypoxemic respiratory failure have poor treatment outcomes. Anemia patients in the ICU had adverse outcomes and a higher extubation failure rate [23,24]. Patients who survived acute respiratory distress syndrome with anemia at ICU discharge were associated with worsened exercise capacity and more activities of daily living dependencies [25]. Keng et al. reported an increased risk of mechanical ventilator weaning failure in patients with anemia and poor oxygenation at respiratory care center admission [26]. Reade et al. demonstrated that anemia patient (Hgb < 10) is associated with 90-day mortality in hospitalized community-acquired pneumonia [27]. Recently, a large, multicenter cohort study reported that COVID-19 patients with anemia are associated with disease severity and mortality [28].”

3 Why there are no inclusive and exclusive criteria?

Response: Thanks for the reviewer’s question. We mentioned the inclusion and exclusive criteria on page 2 in the subsection of the Ethics statement and patients in the Materials and Methods section : “The inclusion criteria were as follows: (1) reported dyspnea without oxygen support, (2) age between 20 and 90 years, (3) meeting the AHRF criteria (SpO2 < 91% or PaO2 < 60 mmHg without oxygen support, PaO2/FiO2 ratio < 300 with oxygen support), and (4) suitable for HFNC therapy as per the attending physician’s discretion. The exclusion criteria included pregnancy, brain injury, epilepsy, myocardial infarction, and missing informed consent (patients who did not agree to participate).”

4 Finally, the authors asserted that the causes of the diseases were heterogeneous, which may have caused false statistical significance in some of the parameters investigated. This is why I find that at this point we still need to carry on a discussion that allows us to bypass this deficit...

Response: Thanks for the reviewer’s essential question. We understood the reviewer’s concern about the causes of the heterogeneous diseases in the general respiratory ward. Because of the cause of acute hypoxemic respiratory failure in multifactor, we mentioned the limitation of this study on page 12 of the manuscript: “Third, the causes of the diseases were heterogeneous, which may have caused statistical insignificance in some of the investigated parameters. Further large-scale studies focusing on a single disease for these prognostic factors are necessary to validate these clinical findings.”

5 I would highlight more the failure rather than the success considering the small number of patients included in this study

Response: Thanks for the reviewer’s important suggestion. We added the reasons of HFNC failure in detail as suggested by the Reviewer in a previous comment. In addition, we discussed the factors that would cause HFNC treatment failure as survival, including anemia, NL ratio, and EIT CoV finding.

Comments on the Quality of English Language

minor editing needed

Response: Thanks for the reviewer’s crucial suggestion. We consulted Editage for English language editing.

Reviewer 2 Report

Comments and Suggestions for Authors

Comments to the Author

The authors showed the potential of EIT for monitoring efficacy of HFNC. This study has clinical significance and novelty. The resolution of the following issues for acceptance of biomedicine.

Major comment

1.      Please show the smoking history, rate of patients with interstitial lung disease, or heart failure, and analyze the association between prognosis and these factors as explanatory variables.

2.      Please provide more details of the respiratory failure caused by lung cancer.

Minor comments

1.      Please unify the axis of characters in the table.

Author Response

Review Report, Reviewer 2

Major comment

  1. Please show the smoking history, rate of patients with interstitial lung disease, or heart failure, and analyze the association between prognosis and these factors as explanatory variables.

Response:  Thanks for the reviewer's crucial suggestion. We added smoking history and patients with heart failure in our revised manuscript. However, no patient with interstitial lung disease was recruited in this study. In Tables 1 and 2, we added factors (smoking status and heart failure) and analyzed the association between these two factors and the outcome of HFNC treatment. These two factors were similar between HFNC success or failure groups. In Table 4, we only added smoking status to analyze its prognostic role in in-hospital survival. We found that smoking status was not a predicting factor for in-hospital survival. We did not include heart failure as a factor in Table 4 because of the minimal patient number of heart failure (n=3) in this study.

  1. Please provide more details of the respiratory failure caused by lung cancer.

Response:  Thanks for the reviewer's essential question. There were four patients with acute hypoxemic respiratory failure caused by advanced lung cancer. We included the following sentences on page 4 in our revised manuscript: "….., an advanced lung cancer patient suffered from pulmonary lymphangitic carcinomatosis with a do-not-intubate order treated with HFNC oxygen therapy for four days but failed. Subsequently, she received non-invasive mechanical ventilator support and passed away after one day.", and  “Twenty-four patients with pneumonia, 5 patients with obstructive lung diseases, and 3 patients with advanced lung cancer discontinued HFNC and converted to conventional oxygen therapy after proper treatments. Among lung cancer cases, 2 patients with malignant pleural effusion successfully removed HFNC after pigtail catheter drainages and albumin plus diuretics treatment; 1 patient who encountered hemoptysis and aspiration was stabilized after medical treatment and weaned off HFNC.”

Minor comments

  1. Please unify the axis of characters in the table.

Response: Thanks for the reviewer’s important suggestion. We unify the axis of characters in Table 1 and 2 in the revised manuscript.

Reviewer 3 Report

Comments and Suggestions for Authors

The authors have done a   prospective observational study enrolling 42 patients due to acute hypoxemic respiratory failure, requiring HFNC oxygen therapy in the general adult respiratory ward.

 They concluded that HFNC in the general respiratory ward may be a potential rescue therapy for patients with respiratory failure and EIT can potentially monitor patients receiving HFNC therapy.

The study is well designed and the Introduction is written well. In introduction they have mentioned about the recent developments on the use of high-flow nasal cannula oxygen in various cases of hypoxemic respiratory failures.

The materials and methods are well described. They have explained the method of administering the HFNC as well as EIT procurement.

The results are discussed well mentioning the success of HFNC as well as the reason of failure for the treatment. EIT was measured for 24 patients out of which 20 patients were successfully treated with HFNC and there were only 4 failures.

Finally, they concluded that HFNC therapy was clinically applicable for the management of patients with acute hypoxic respiratory failure in a general respiratory ward. The success rate of weaning off HFNC was 76%. The in-hospital mortality rate was 29%. EIT can potentially help monitor patients receiving HFNC therapy. I have a few general comments:

In page 2 line #80, “……HFNC therapy per the attending physician’s discretion ….” should be written as “HFNC therapy as per the attending physician’s discretion.”

The following paper should be cited:

Koyauchi T, Hasegawa H, Kanata K, Kakutani T, Amano Y, Ozawa Y, Matsui T, Yokomura K, Suda T. Efficacy and tolerability of high-flow nasal cannula oxygen therapy for hypoxemic respiratory failure in patients with interstitial lung disease with do-not-intubate orders: a retrospective single-center study. Respiration. 2018 Oct 17;96(4):323-9.

The paper is well written and I recommend it for publication with a minor revision.

Comments on the Quality of English Language

A few mistakes in Grammar are present.

In page 2 line #80, “……HFNC therapy per the attending physician’s discretion ….” should be written as “HFNC therapy as per the attending physician’s discretion.”

Author Response

Review Report, Reviewer 3

Reviewer 3:

I have a few general comments:

  1. In page 2 line #80, “……HFNC therapy per the attending physician’s discretion ….” should be written as “HFNC therapy as per the attending physician’s discretion.”

Response:  Thanks for the reviewer’s important suggestion. We corrected this sentence as “HFNC therapy as per the attending physician’s discretion.”

  1. The following paper should be cited:

Koyauchi T, Hasegawa H, Kanata K, Kakutani T, Amano Y, Ozawa Y, Matsui T, Yokomura K, Suda T. Efficacy and tolerability of high-flow nasal cannula oxygen therapy for hypoxemic respiratory failure in patients with interstitial lung disease with do-not-intubate orders: a retrospective single-center study. Respiration. 2018 Oct 17;96(4):323-9.

Response: Thanks for the reviewer’s essential suggestion. We added the sentence “In addition, HFNC therapy improves respiratory rate, tolerance, and comfort in interstitial lung disease patients with acute hypoxemic respiratory failure who receive supportive care [39].” on pages 12. We cited this paper as “reference 39” in our revised manuscript.

Round 2

Reviewer 1 Report

Comments and Suggestions for Authors

Overall my opinion about this manuscript is positive after the corrections

Cogratulations

Author Response

Comments and Suggestions for Authors

Overall my opinion about this manuscript is positive after the corrections

Congratulations

Response: Thanks for reviewer’s positive comment. We appreciate the reviewer’s essential and helpful suggestions in the review process.

Reviewer 2 Report

Comments and Suggestions for Authors

Author almost resonded to my comment correctly. Please show that no patients with ILD was included in study subjects.

Author Response

Comments and Suggestions for Authors

Author almost responded to my comment correctly. Please show that no patients with ILD was included in study 

Response:  Thanks for the reviewer's crucial suggestion. On page 4 of the revised manuscript, we added, “No patient with interstitial lung disease was included in the study subjects.” We appreciate the reviewer’s essential and helpful suggestions in the review process.